# Vocabulary for Universal Approximation: A Linguistic Perspective of Mapping Compositions

## Abstract

In recent years, deep learning-based sequence modelings, such as language models, have received much attention and success, which pushes researchers to explore the possibility of transforming non-sequential problems into a sequential form. Following this thought, deep neural networks can be represented as composite functions of a sequence of mappings, linear or nonlinear, where each composition can be viewed as a *word*. However, the weights of linear mappings are undetermined and hence require an infinite number of words. In this article, we investigate the finite case and constructively prove the existence of a finite *vocabulary* $V = \{\phi_i : \mathbb{R}^d \to \mathbb{R}^d | i = 1, ..., n\}$ with $n = O(d^2)$ for the universal approximation. That is, for any continuous mapping $f : \mathbb{R}^d \to \mathbb{R}^d$, compact domain $\Omega$ and $\varepsilon > 0$, there is a sequence of mappings $\phi_{i_1}, ..., \phi_{i_m} \in V, m \in \mathbb{Z}_+$, such that the composition $\phi_{i_m} \circ ... \circ \phi_{i_1}$ approximates $f$ on $\Omega$ with an error less than $\varepsilon$. Our results demonstrate an unusual approximation power of mapping compositions which is a little similar to the compositionality in linguistics which is the idea that a finite vocabulary of basic elements can be combined via grammar to express an infinite range of meanings.

## 1 Introduction

Cognitive psychologists and linguisticians have long recognized the importance of languages (Pinker, 2003), which has been further highlighted by the popularity of language models such as BERT (Devlin et al., 2018) and GPT (Brown et al., 2020). These models, based on RNNs or Transformers, have revolutionized natural language processing by transforming it into a sequence learning problem. They can handle the long-term dependencies in text and generate coherent text based on the previous content, making them invaluable tools in language understanding and generation (Vaswani et al., 2017). The success of these models has also led to a new approach to solving non-sequential problems by transforming them into sequential ones. For instance, image processing can be turned into a sequence learning problem by segmenting an image into small blocks, arranging them in a certain order, and then processing the resulting sequence using sequence learning algorithms to achieve image recognition (Dosovitskiy et al., 2021). The use of sequence learning algorithms has also been extended to reinforcement learning (Chen et al., 2021), such as the decision transformer outputs the optimal actions by leveraging a causally masked transformer and exceeds the state-of-the-art performance.

Sequence modeling has opened up new possibilities for solving a wide range of problems, and this trend seems to hold in the field of theoretical research. As is well known, artificial neural networks have universal approximation capabilities, and wide or deep feedforward networks can approximate continuous functions on a compact domain arbitrarily well (Cybenkot, 1989; Hornik et al., 1989; Leshno et al., 1993). However, in practical applications such as AlphaFold (Jumper et al., 2021), BERT (Devlin et al., 2018) and GPT (Brown et al., 2020), the residual network structures (He et al., 2016a;b) are more preferred than the feedforward structures. It is observed that residual networks (ResNets) are forward Euler discretizations of dynamical systems (E, 2017; Sander et al., 2022), and this relationship has spawned a series of dynamical system-based neural network structures such as the neural ODE (Chen et al., 2018). The dynamical system-based neural network structures are expected to play an important role in various fields.

Notably, both the language models and the dynamical systems are linked to time series modeling and have been effectively applied to non-sequential problems. This observation naturally leads us to question: *is there an intricate relationship between their individual successes?* This article aims to ponder upon the question. Through a comparative study, we obtain some initial results from the perspective of universal approximation. Specifically, we demonstrate that there exists a finite set of mappings, referred to as the *vocabulary $V$*, choosing as flow maps of some autonomous dynamical system $x'(t) = f(x(t))$, such that any continuous mapping can be approximated by compositing a sequence of mappings in the vocabulary $V$. This bears a resemblance to the way complex information is conveyed in natural language through constructing phrases, sentences, and ultimately paragraphs and compositions. Table 1 provides an intuitive representation of this similarity.

Table 1: Comparison of natural languages and dynamical systems in dimension $d$

|  | **English** | **Flow map of dynamical systems** [‡] |
| --- | --- | --- |
| Vocabulary | $\sim$140,000[†] | $O(d^2)$ |
| Word | I, you, am, is, are, apple, banana, car, buy, do, have, blue, red,... | $\phi^\tau_{\pm e_1}, \phi^\tau_{\pm e_2}, ..., \phi^\tau_{\pm e_d}, \phi^\tau_{\pm E_{11}x}, \phi^\tau_{\pm E_{12}x}, \phi^\tau_{\pm E_{21}x}, ..., \phi^\tau_{\pm E_{dd}x}, \phi^\tau_{\pm \text{ReLU}(x)}, \cdots$ |
| Phrase | A big deal, easier said than done, time waits for no man, ... | $\phi^\tau_{e_1} \bullet \phi^\tau_{-e_2}, \phi^\tau_{e_1} \bullet \phi^\tau_{E_{11}x} \bullet \phi^\tau_{\text{ReLU}(x)}, \cdots$ |
| Sentence | It was the best of times, it was the worst of times, it was the age of wisdom, it was the age of foolishness, ... | $\phi^\tau_{e_3} \bullet \phi^\tau_{\text{ReLU}(x)} \bullet \phi^\tau_{-E_{21}x} \bullet \phi^\tau_{\text{ReLU}(x)} \bullet \phi^\tau_{E_{23}x} \bullet \phi^\tau_{-e_2} \bullet \phi^\tau_{\text{ReLU}(x)} \bullet \phi^\tau_{E_{11}x} \bullet \phi^\tau_{e_1} \bullet \cdots$ |

† The number of words, phrases, and meanings in Cambridge Advanced Learner's Dictionary.
‡ Notations are provided in Section 2.

## 1.1 CONTRIBUTIONS

1. We proved that it is possible to achieve the universal approximation property by compositing a sequence of mappings in a finite set $V$. (Theorem 2.2 and Corollary 2.3).

2. Our proof is constructive as we designed such a $V$ that contains a finite number of flow maps of dynamical systems. (Theorem 2.6)

## 1.2 RELATED WORKS

**Universal approximation.** The approximation properties of neural networks have been extensively studied, with previous studies focusing on the approximation properties of network structures such as feedforward neural networks (Cybenkot, 1989; Hornik et al., 1989; Leshno et al., 1993) and residual networks (He et al., 2016a;b). In these networks, the structure is fixed and the weights are adjusted to approximate target functions. Although this paper also considers universal approximation properties, we use a completely different way. We use a finite set of mappings, and the universal approximation is achieved by composing sequences of these mappings. The length of the mapping sequence is variable, which is similar to networks with a fixed width and variable depth (Lu et al., 2017; Johnson, 2019; Kidger & Lyons, 2020; Park et al., 2021; Beise & Da Cruz, 2020; Cai, 2022). However, in our study, we do not consider learnable weights; instead, we consider the composition sequence, which is different from previous research.

**Residual network, neural ODE, and control theory.** The word mapping constructed in this paper is partially based on the numerical discretization of dynamical systems and therefore has a relationship with residual networks and neural ODEs. Residual networks (He et al., 2016a;b) are currently one of the most popular network structures and can be viewed as a forward Euler discretization of neural ODEs (Chen et al., 2018). Recently, Li et al. (2022) and Tabuada & Gharesifard (2020; 2022) studied the approximation properties of neural ODEs. Their basic idea is employing controllability results in control theory to construct source terms that approximate a given finite number of input-output pairs, thus obtaining the approximation properties of functions in the $L^p$ norm or continuous norm sense. Additionally, Duan et al. (2022) proposed an operator splitting format that

discretizes neural ODEs into Leaky-ReLU fully connected networks. Partially inspired by Duan et al.'s construction, we designed a special splitting method to finish Part 1 of our construction.

It's worth noting that all neural networks mentioned above can be represented as compositions of mapping sequences. However, the networks involve an infinite number of mappings, which is different from our construction which only requires a finite number of mappings.

**Compositionality.** Our results demonstrate that the composition is a powerful operator that allows us to achieve the universal approximation property on compact domains by using a finite number of mappings. This is a little similar to the concept of compositionality in linguistics, especially in the Montagovian framing (Montague, 1970; Kracht, 2012), which is the idea that a finite vocabulary of basic elements can be combined via a grammar to express an infinite range of meanings. Recently, researchers have explored the capabilities of neural models to acquire compositionality while learning from data (Dankers et al., 2022; Valvoda et al., 2022). However, they focused on algebraic relations rather than approximations. It's interesting to think whether these studies and ours can be connected.

**Word embeding.** The finite mapping vocabulary might be related to the word embedding in natural language processing. The most basic model involves embedding words as vectors and then summing these word vectors to obtain the sentence vector (Mikolov et al., 2013). However, the summation operator is commutative, and thus vector embedding models fail to capture any notion of word order. To address this limitation, Rudolph & Giesbrecht (2010) proposed modeling words as matrices rather than vectors and composing sentence embeddings through matrix multiplication instead of addition. For recent advancements in this direction, we refer to Mai et al. (2018); Asaadi et al. (2023). It is noteworthy that, to the best of our knowledge, prior research in this domain has not delved into the approximation properties. Leveraging the techniques presented in this paper, we can readily establish the existence of a finite vocabulary for both vector embedding and matrix embedding (refer to Appendix D). Furthermore, it is important to note that vector space and matrix space are finite-dimensional, while the continuous function space is infinite-dimensional. This suggests that embedding words as nonlinear mappings could enhance the expressiveness of sentences. However, there is limited exploration in this direction.

## 1.3 OUTLINE

We state the main result for universal approximation in Section 2, which includes notations, main theorems, and ideas for construction and proof. Before providing the detailed construction in Section 4, we add a Section 3 to introduce flow maps and the techniques we used. Finally, in Section 5 we discuss the result of this paper. All formal proof of the theorems is provided in the Appendix.

## 2 NOTATIONS AND MAIN RESULTS

### 2.1 PRELIMINARIES

The statement and the proof of our main results contain some concepts in mathematics. Here we provide a brief introduction for them, which is enough to understand most parts of this paper.

One concept is the orientation-preserving (OP) diffeomorphisms of $\mathbb{R}^d$. A differentiable map $f : \mathbb{R}^d \to \mathbb{R}^d$ is called a diffeomorphism if it is a bijection and its inverse $f^{-1}$ is differentiable as well. In addition, a diffeomorphism $f$ of $\mathbb{R}^d$ is called orientation-preserving if the Jacobian of $f$ is positive everywhere. A simple example of OP diffeomorphisms is the linear map $f : x \to Px$ where $x \in \mathbb{R}^d$ and $P$ is a square matrix with positive determinant.

Another concept is the flow map of dynamical systems. Here the dynamical system is characterized by the following ordinary differential equation (ODE) in dimension $d$,

$$\begin{cases} \dot{x}(t) = v(x(t), t), t \in (0, \tau), \\ x(0) = x_0 \in \mathbb{R}^d, \end{cases} \tag{1}$$

where $v : \mathbb{R}^d \to \mathbb{R}^d$ is the velocity field and $x_0$ is the initial value. When the field $v$ satisfies some conditions, such as Lipschitz continuous, the ODE (1) has a unique solution $x(t), t \in [0, \tau]$. Then the map from the initial state $x_0$ to $x(\tau)$, the state of the system after time $\tau$, is called the flow map

and denoted by $\phi^\tau_{v(x,t)}(x_0)$, where $x_0$ is allowed to vary. A basic property is that the flow maps are naturally OP diffeomorphisms. For example, let $A$ be a square matrix and $v(x,t) = Ax$, then the flow map $\phi^\tau_{v(x,t)}(x_0)$ is a linear map $\phi^\tau_{Ax}(x_0) = e^{A\tau}x_0$, where $e^{A\tau}$ is the matrix exponential of $A\tau$. A deeper introduction and understanding of flows and dynamical systems can be found in Chapter 1 of Arrowsmith & Place (1990).

## 2.2 NOTATIONS

For a (vector valued) function class $\mathcal{F}$, the *vocabulary* $V$ is defined as a finite subset of $\mathcal{F}$, *i.e.*,

$$V = \{\phi_1, \phi_2, ..., \phi_n\} \subset \mathcal{F}, \quad n \in \mathbb{Z}_+. \tag{2}$$

Each $\phi_i \in V$ is called a *word*. We will consider a sequence of functions, $\phi_{i_1}, \phi_{i_2}, ..., \phi_{i_m} \in V$, and their composition, called as a *sentence*, to generate the hypothesis function space,

$$\mathcal{H}_V = \{\phi_{i_1} \bullet \phi_{i_2} \bullet ... \bullet \phi_{i_m} | \phi_{i_1}, \phi_{i_2}, ..., \phi_{i_m} \in V, m \in \mathbb{Z}_+\}. \tag{3}$$

Particularly, some (short) sentences are called *phrases* for some purpose. Here the operator $\bullet$ is defined as function composition from left to right, which aligns the composition order to the writing order, *i.e.*

$$\phi_{i_1} \bullet \phi_{i_2} \bullet ... \bullet \phi_{i_m} = \phi_{i_m} \circ ... \circ \phi_{i_2} \circ \phi_{i_1} = \phi_{i_m}(...(\phi_{i_2}(\phi_{i_1}(\cdot)))...). \tag{4}$$

In additional, we use $\phi^{\bullet m}$, to denote the mapping that composites $\phi$ $m$ times.

In this paper, we consider two function classes: (1) $C(\mathbb{R}^d, \mathbb{R}^d)$, continuous functions from $\mathbb{R}^d$ to $\mathbb{R}^d$, (2) $\text{Diff}_0(\mathbb{R}^d)$, OP diffeomorphisms of $\mathbb{R}^d$. Particularly, we will restrict the functions on a compact domain $\Omega \subset \mathbb{R}^d$ and define the universal approximation property as below.

**Definition 2.1** (Universal approximation property, UAP). *For the compact domain $\Omega$ in dimension $d$, the target function space $\mathcal{F}$ and the hypothesis space $\mathcal{H}$, we say*

    *1. $\mathcal{H}$ has C-UAP for $\mathcal{F}$, if for any $f \in \mathcal{F}$ and $\varepsilon > 0$, there is a function $h \in \mathcal{H}$ such that*

$$\|f(x) - h(x)\| < \varepsilon, \quad \forall x \in \Omega.$$

    *2. $\mathcal{H}$ has $L^p$-UAP for $\mathcal{F}$, if for any $f \in \mathcal{F}$ and $\varepsilon > 0$, there is a function $h \in \mathcal{H}$ such that*

$$\|f - h\|_{L^p(\Omega)} = \left( \int_\Omega |f(x) - h(x)|^p dx \right)^{1/p} < \varepsilon, \quad p \in [1, +\infty).$$

## 2.3 MAIN THEOREM

Our main result is Theorem 2.2 and its Corollary 2.3 which show the existence of a finite function vocabulary $V$ for the universal approximation property.

**Theorem 2.2.** *Let $\Omega \subset \mathbb{R}^d$ be a compact domain. Then, there is a finite set $V \subset \text{Diff}_0(\mathbb{R}^d)$ such that the hypothesis space $\mathcal{H}_V$ in Eq. (3) has C-UAP for $\text{Diff}_0(\mathbb{R}^d)$.*

**Corollary 2.3.** *Let $\Omega \subset \mathbb{R}^d$ be a compact domain, $d \geq 2$ and $p \in [1, +\infty)$. Then, there is a finite set $V \subset C(\mathbb{R}^d, \mathbb{R}^d)$ such that the hypothesis space $\mathcal{H}_V$ in Eq. (3) has $L^p$-UAP for $C(\mathbb{R}^d, \mathbb{R}^d)$.*

The Corollary 2.3 is based on the fact that OP diffeomorphisms can approximate continuous functions under the $L^p$ norm provided the dimension is larger than two (Brenier & Gangbo, 2003) . Next, we only need to prove Theorem 2.2.

**Remark 2.4.** *We are considering functions to have the same dimension of the input and output, for simplicity. Our results can be directly extended to the case of different input and output dimensions. In fact, for $f \in C(\mathbb{R}^{d_x}, \mathbb{R}^{d_y})$, one can lift it as a function $\tilde{f} \in C(\mathbb{R}^d, \mathbb{R}^d)$ with some $d \geq \max(d_x, d_y)$. For example, let $f = A_{in} \bullet \tilde{f} \bullet A_{out}$ where $A_{in} \in C(\mathbb{R}^{d_x}, \mathbb{R}^d)$ and $A_{out} \in C(\mathbb{R}^d, \mathbb{R}^{d_y})$ are two fixed affine mappings.*

## 2.4 SKETCH OF THE PROOF

Our proof for Theorem 2.2 is constructive, by concerning the flow maps of ODEs. In particular, our construction will use the following class of candidate flow maps in dimension $d$,

$$H_1 = \left\{ \phi_{Ax+b}^\tau \mid A \in \mathbb{R}^{d \times d}, b \in \mathbb{R}^d, \tau \geq 0 \right\} \equiv \left\{ \phi : x \to e^{\tilde{A}} x + \tilde{b} \mid \tilde{A} \in \mathbb{R}^{d \times d}, \tilde{b} \in \mathbb{R}^d \right\}, \quad (5)$$

$$H_2 = \left\{ \phi_{\Sigma_{\boldsymbol{\alpha}, \boldsymbol{\beta}}(x)}^\tau \mid \boldsymbol{\alpha}, \boldsymbol{\beta} \in \mathbb{R}^d, \tau \geq 0 \right\} \equiv \left\{ \phi : x \to \Sigma_{\tilde{\boldsymbol{\alpha}}, \tilde{\boldsymbol{\beta}}}(x) \mid \tilde{\boldsymbol{\alpha}}, \tilde{\boldsymbol{\beta}} \in (0, +\infty)^d \right\}, \quad (6)$$

where $\Sigma_{\boldsymbol{\alpha}, \boldsymbol{\beta}}$ is the generalized leaky-ReLU functions defined as below. We say $H_1$ the affine flows and $H_2$ the leaky-ReLU flows.

**Definition 2.5** (Generalized leaky-ReLU). *Define the generalized leaky-ReLU function as $\sigma_{\alpha, \beta}$ : $\mathbb{R} \to \mathbb{R}$ and $\Sigma_{\boldsymbol{\alpha}, \boldsymbol{\beta}} : \mathbb{R}^d \to \mathbb{R}^d$, with $\alpha, \beta \in \mathbb{R}$, $\boldsymbol{\alpha} = (\alpha_1, ..., \alpha_d) \in \mathbb{R}^d$, $\boldsymbol{\beta} = (\beta_1, ..., \beta_d) \in \mathbb{R}^d$,*

$$\sigma_{\alpha, \beta}(x) = \begin{cases} \alpha x, & x < 0 \\ \beta x, & x \geq 0 \end{cases}, \quad \Sigma_{\boldsymbol{\alpha}, \boldsymbol{\beta}}(x) = \left( \sigma_{\alpha_1, \beta_1}(x_1), ..., \sigma_{\alpha_d, \beta_d}(x_d) \right). \quad (7)$$

Generalized leaky-ReLU functions are piecewise linear functions. Using this notation, the traditional ReLU and leaky-ReLU functions are $\text{ReLU}(x) \equiv \sigma_0(x) \equiv \sigma_{0,1}(x)$ and $\sigma_\alpha(x) \equiv \sigma_{\alpha,1}(x)$ with $\alpha \in (0, 1)$, respectively. For vector input $x$, we use $\sigma_{\alpha, \beta}$ as an equivilant notation of $\Sigma_{\alpha \mathbf{1}, \beta \mathbf{1}}$.

We will show that the following set $V$ meets our requirement for universal approximations,

$$V = \left\{ \phi_{\pm e_i}^\tau, \phi_{\pm E_{ij} x}^\tau, \phi_{\pm \Sigma_{e_i, 0}(x)}^\tau, \phi_{\pm \Sigma_{0, e_i}(x)}^\tau \mid i, j \in \{1, 2, ..., d\}, \tau \in \{1, \sqrt{2}\} \right\}, \quad (8)$$

where $e_i \in \mathbb{R}^d$ is the $i$-th unit coordinate vector, $E_{ij}$ is the $d \times d$ matrix that has zeros in all entries except for a 1 at the index $(i, j)$. Obviously, $V \subset \text{Diff}_0(\mathbb{R}^d)$ is a finite set with $O(d^2)$ functions.

**Theorem 2.6.** *Let $\Psi \in Diff_0(\Omega)$ be an orientation preserving diffeomorphism, $\Omega$ be a compact domain $\Omega \subset \mathbb{R}^d$. Then, for any $\varepsilon > 0$, there is a sequence of flow maps, $\phi_1, \phi_2, ..., \phi_n \in V, n \in \mathbb{Z}_+$, such that*

$$\|\Psi(x) - (\phi_1 \bullet \phi_2 \bullet ... \bullet \phi_n)(x)\| \leq \varepsilon, \quad \forall x \in \Omega. \quad (9)$$

Theorem 2.6 provides a constructive proof for Theorem 2.3. The proof of Theorem 2.6 can be separated into the following two parts.

**Part 1**: Approximate each flow map in $H_1$ and $H_2$ by compositing a sequence of flow maps in $V$.

**Part 2**: Approximate $\Psi \in \text{Diff}_0(\mathbb{R}^d)$ by compositing a sequence of flow maps in $H_1 \cup H_2$. Particularly, we approximate $\Psi$ by $g_L$ of the form

$$g_L = h_0 \bullet h_1^* \bullet h_1 \bullet h_2^* \bullet h_2 \bullet ... \bullet h_L^* \bullet h_L, \quad h_i \in H_1, h_i^* \in H_2, L \in \mathbb{Z}_+. \quad (10)$$

The validation of such constructed $V$ is technical and will be proved in Section 3 and Section 4. Here we only explain the main ideas. First of all, we note that to approximate a composition map $T$, we only need to approximate each component in $T$, which is detailed in the following Lemma 2.7.

**Lemma 2.7.** *Let map $T = F_1 \bullet ... \bullet F_n$ be a composition of $n$ continuous functions $F_i$ defined on an open domain $D_i$, and let $\mathcal{F}$ be a continuous function class that can uniformly approximate each $F_i$ on any compact domain $\mathcal{K}_i \subset D_i$. Then, for any compact domain $\mathcal{K} \subset D_1$ and $\varepsilon > 0$, there are $n$ functions $\tilde{F}_1, ..., \tilde{F}_n$ in $\mathcal{F}$ such that*

$$\|T(x) - \tilde{F}_1 \bullet ... \bullet \tilde{F}_n(x)\| \leq \varepsilon, \quad \forall x \in \mathcal{K}. \quad (11)$$

For Part 1, the validation involves three techniques in math: the Lie product formula (Hall, 2015), the splitting method (Holden et al., 2010) and the Kronecker's theorem (Apostol, 1990). We take $\phi_b^1 \in H_1, b = \sum_{i=1}^d \beta_i e_i, \beta_i \geq 0$, as an example to illustrate the main idea. Firstly, motivated by the Lie product formula or the splitting method, we can approximate $\phi_b^1$ by

$$\phi_b^1 \approx \left( \phi_{e_1}^{\beta_1/n} \bullet \phi_{e_2}^{\beta_2/n} \bullet ... \bullet \phi_{e_d}^{\beta_d/n} \right)^{\bullet n}, \quad n \in \mathbb{Z}_+, \quad (12)$$

with $n$ large enough. Secondly, each $\phi_{e_i}^{\beta_i/n}$ can be approximated by

$$\phi_{e_i}^{\beta_i/n} \approx (\phi_{e_i}^1)^{\bullet p_i} \bullet (\phi_{-e_i}^{\sqrt{2}})^{\bullet q_i} \in \mathcal{H}_V, \quad p_i, q_i \in \mathbb{Z}_+ \quad (13)$$

where $p_i$ and $q_i$ are non-negative integers such that $|p_i - q_i\sqrt{2} - \beta_i/n|$ is small enough according to the Kronecker's theorem (Apostol, 1990) as $\sqrt{2}$ is an irrational number. Finally, $\phi_b^1$ can be approximated by compositing a sequence of flow maps in $V$. The case for $\phi_{Ax+b}^\tau$ and $\phi_{\Sigma_{\boldsymbol{\alpha},\boldsymbol{\beta}(x)}}^\tau$ in $H_1$ and $H_2$ can be done in the same spirit.

Then for Part 2, we note that the $g_L$ we constructed in Eq. (10) is similar to a feedforward neural network $g_L$ with width $d$ and depth $L$. The form of $g_L$ is motivated by a recent work of Duan et al. (2022) which proved that vanilla feedforward leaky-ReLU networks with width $d$ can be a discretization of dynamic systems in dimension $d$. However, affine transformations in general networks are not necessarily OP diffeomorphisms, and one novelty of this paper is improving the technique to construct $P_i$ as flow maps. Importantly, making them flow maps helps with employing the construction in Part 1.

# 3 PROOF OF THE CONSTRUCTION PART 1

To warm up, we show some flow maps of autonomous ODEs below,

$$\dot{x}(t) = b, \quad x(0) = x_0 \qquad \Rightarrow \qquad x(t) = \phi_b^t(x_0) = x_0 + bt, \tag{14}$$

$$\dot{x}(t) = Ax(t) + b, \quad x(0) = x_0 \qquad \Rightarrow \qquad x(t) = e^{At}x_0 + \int_0^t e^{A(t-\tau)}bd\tau, \tag{15}$$

$$\dot{x}(t) = a\sigma_0(x(t)), \quad x(0) = x_0 \qquad \Rightarrow \qquad x(t) = \phi_{a\sigma_0(x)}^t(x_0) = e^{at}\sigma_{e^{-at}}(x_0), \tag{16}$$

$$\dot{x}(t) = a\sigma_0(-x(t)), \quad x(0) = x_0 \qquad \Rightarrow \qquad x(t) = \phi_{a\sigma_0(-x)}^t(x_0) = \sigma_{e^{-at}}(x_0). \tag{17}$$

Here $\sigma_0$ and $\sigma_{e^{-at}}$ are ReLU and leaky-ReLU functions, respectively. Next, we provide some properties to verify a given map to be an affine flow map in $H_1$ or a leaky-ReLU flow map in $H_2$.

## 3.1 AFFINE FLOWS AND LEAKY-RELU FLOWS

Consider the affine transformation $P : x \to Wx + b$ and examine conditions of $P$ to be a flow map. Generally, if $W$ is nonsingular and has real matrix logarithm $\ln(W)$, then $P$ is an affine flow map, as we can represent $P$ as $P(x) = Wx + b = \phi_{Ax+\tilde{b}}^1$ where $A = \ln(W)$ and $\tilde{b} = \int_0^1 e^{A(\tau-1)}bd\tau$. As it is hard to verify $\ln(W)$ is a real matrix (Culver, 1966), we are happy to construct some special matrix $W$. The following properties are useful.

**Proposition 3.1.** *(1) Let $Q$ be a nonsingular matrix. If $x \to Wx$ is an affine flow map then the map $x \to QWQ^{-1}x$, $x \to W^T x$ and $x \to W^{-1}x$ also are. (2) Let $U$ be an upper triangular matrix below with $\lambda > 0$, then the map $x \to Ux$ is an affine flow map for arbitrary vector $w_{2:d}$,*

$$U = \begin{pmatrix} \lambda & w_{2:d} \\ 0 & I_{d-1} \end{pmatrix}. \tag{18}$$

Here $I_{d-1}$ is the $(d-1)$th order identity matrix. The property (1) is because $\ln(QWQ^{-1}) = Q\ln(W)Q^{-1}$ and $\ln(W^T) = \ln(W)^T$. The property (2) can be obtained by employing the formula,

$$\ln \begin{pmatrix} \lambda & w_{2:d} \\ 0 & I_{d-1} \end{pmatrix} = \begin{pmatrix} \ln(\lambda) & \frac{\ln(\lambda)}{\lambda-1}w_{2:d} \\ 0 & 0 \end{pmatrix}, \quad \lambda \neq 1. \tag{19}$$

When $\lambda = 1$, the formula is simplified as $\ln(U) = U - I_d$.

Next, we consider the leaky-ReLU flow maps.

By directly calculate the flow map $\phi_{\Sigma_{\boldsymbol{\alpha},\boldsymbol{\beta}(x)}}^\tau$ with $\boldsymbol{\alpha}, \boldsymbol{\beta} \in \mathbb{R}^d$, we have

$$\phi_{\Sigma_{\boldsymbol{\alpha},\boldsymbol{\beta}(x)}}^\tau(x) = \Sigma_{\tilde{\boldsymbol{\alpha}},\tilde{\boldsymbol{\beta}}}(x), \tag{20}$$

where $\tilde{\boldsymbol{\alpha}} = (e^{\tau\alpha_1}, ..., e^{\tau\alpha_d})$ and $\tilde{\boldsymbol{\beta}} = (e^{\tau\beta_1}, ..., e^{\tau\beta_d})$. The following property is implied.

**Proposition 3.2.** *If $\tilde{\boldsymbol{\alpha}}, \tilde{\boldsymbol{\beta}} \in (0,\infty)^d$, then the map $\Sigma_{\tilde{\boldsymbol{\alpha}},\tilde{\boldsymbol{\beta}}}$ is a leaky-ReLU flow map.*

## 3.2 Application of Lie product formula

**Theorem 3.3** (Lie product formula). *For all matrix $A, B \in \mathbb{R}^{d \times d}$, we have*

$$e^{A+B} = \lim_{n \to \infty} \left( e^{A/n} e^{B/n} \right)^n = \lim_{n \to \infty} \left( \phi_{Ax}^{1/n} \bullet \phi_{Bx}^{1/n} \right)^{\bullet n} \tag{21}$$

Here $e^A$ denotes the matrix exponential of $A$, which is also the flow map $\phi_{Ax}^1$ of the autonomous system $x'(t) = Ax(t)$. The proof can be found in Hall (2015) for example and the formula can be extended to multi-component cases. The formula can also be derived from the operator splitting approach (Holden et al., 2010), which allows us to obtain the following result.

**Lemma 3.4.** *Let $v_i : \mathbb{R}^d \to \mathbb{R}^d, i = 1, 2, ..., m$ be Lipschitz continuous funcitons, $v = \sum_{i=1}^m v_i$, $\Omega$ be a compact domain. For any $t > 0$ and $\varepsilon > 0$, there is a positive integers $n$, such that the flow map $\phi_v^t$ can be approximated by composition of flow maps $\phi_{v_i}^{t/n}$ , i.e.*

$$\|\phi_v^t(x) - \left( \phi_{v_1}^{t/n} \bullet \phi_{v_2}^{t/n} \bullet ... \bullet \phi_{v_m}^{t/n} \right)^{\bullet n}(x)\| < \varepsilon, \quad \forall x \in \Omega. \tag{22}$$

## 3.3 Application of Kronecker's theorem

**Theorem 3.5** (Kronecker's approximation theorem (Apostol, 1990)). *Let $\gamma \in \mathbb{R}$ be an irrational number, then for any $t \in \mathbb{R}$ and $\varepsilon > 0$, there exist two integers $p$ and $q$ with $q > 0$, such that $|\gamma q + p - t| < \varepsilon$.*

Although Kronecker's Theorem 3.5 is proposed for approximating real numbers, we can employ it in the scenario of approximating the flow map $\phi_v^t$ as it contains a real-time parameter $t$. Choosing $\gamma = -\sqrt{2}$, approximating $t$ by $p - q\sqrt{2}$, then we can approximate $\phi_v^t$ by $\phi_v^{p-q\sqrt{2}}$. Considering positive $t$, we have $p$ is positive as $q$ is. Then the property of flow maps,

$$\phi_v^{p-q\sqrt{2}} = \phi_v^p \bullet \phi_v^{-q\sqrt{2}} = \phi_v^p \bullet \phi_{-v}^{q\sqrt{2}} = (\phi_v^1)^{\bullet p} \bullet (\phi_{-v}^{\sqrt{2}})^{\bullet q}, \tag{23}$$

allow us to prove the following result.

**Lemma 3.6.** *Let $v : \mathbb{R}^d \to \mathbb{R}^d$ be a Lipschitz continuous function, $\Omega$ be a compact domain. For any $t > 0$ and $\varepsilon > 0$, there exist two positive integers $p$ and $q$, such that the flow map $\phi_v^t$ can be approximated by $(\phi_v^1)^{\bullet p} \bullet (\phi_{-v}^{\sqrt{2}})^{\bullet q}$, i.e.*

$$\|\phi_v^t(x) - (\phi_v^1)^{\bullet p} \bullet (\phi_{-v}^{\sqrt{2}})^{\bullet q}(x)\| < \varepsilon, \quad \forall x \in \Omega. \tag{24}$$

**Corollary 3.7.** *For any flow maps $h$ in $H_1 \cup H_2$, $\varepsilon > 0$ and compact domain $\Omega \subset \mathbb{R}^d$, there is a sequence $\phi_1, \phi_2, ..., \phi_m$ in $V$ (Eq. 8) such that*

$$\|h(x) - (\phi_1 \bullet \phi_2 ... \bullet \phi_m)(x)\| < \varepsilon, \quad \forall x \in \Omega. \tag{25}$$

The result is obtained by directly employing Lemma 3.4 and Lemma 3.6 with the following splittings,

$$Ax + b = \sum_{i=1}^d \sum_{j=1}^d a_{ij} E_{ij} x + \sum_{i=1}^d b_i e_i, \quad \Sigma_{\boldsymbol{\alpha}, \boldsymbol{\beta}}(x) = \sum_{i=1}^d \alpha_i \Sigma_{e_i, 0}(x) + \sum_{i=1}^d \beta_i \Sigma_{0, e_i}(x). \tag{26}$$

# 4 Proof of the construction Part 2

This section provides the construction that OP diffeomorphisms can be approximated by compositing a sequence of flow maps in $H_1 \cup H_2$. The construction contains three steps: (1) approximate OP diffeomorphisms by deep compositions using the splitting approach, (2) approximate each splitting component by compositing flow maps in $H_1 \cup H_2$, (3) combine results to finish the construction.

## 4.1 Approximate the OP diffeomorphism by deep compositions

Employing results of Agrachev & Caponigro (2010) and Caponigro (2011), any OP diffeomorphism $\Psi$ can be approximated by flow maps of ODEs. Particularly, we can choose the ODEs as neural

ODEs of the form

$$x'(t) = v(x(t), t) = \sum_{i=1}^{N} s_i(t)\sigma(w_i(t) \cdot x(t) + b_i(t)), \tag{27}$$

where the field function $v$ is a neural network with $N$ hidden neurons, the activation is chosen as the leaky-ReLU function $\sigma = \sigma_\alpha$ for some $\alpha \in (0, 1)$, $s_i \in \mathbb{R}^d$, $w_i \in \mathbb{R}^d$ and $b_i \in \mathbb{R}$ are piecewise smooth functions of $t$. The universal approximation property of neural networks (Cybenko, 1989) implies that $\Psi$ can be approximated by the flow map $\phi_v^\tau$ of Eq. (27) for some $\tau > 0$ and $N \in \mathbb{Z}_+$ big enough.

Following the approach of Duan et al. (2022), we employ a proper splitting numerical scheme to discretize the neural ODE (27). Split the field $v$ as a summation of $Nd$ functions, $v(x, t) = \sum_{i=1}^{N} \sum_{j=1}^{d} v_{ij}(x, t)e_j$, where $e_j$ is the $j$-th axis unit vector and $v_{ij}(x, t) = s_{ij}(t)\sigma(w_i(t) \cdot x + b_i(t))$ are scalar functions. Then the numerical analysis theory of splitting methods (Holden et al., 2010) ensures that the following composition $\Phi$ can approximate $\phi^\tau$ provided the time step $\Delta t := \tau/n$ is sufficiently small,

$$\Phi = T_1 \bullet \cdots \bullet T_n \equiv (T_1^{(1,1)} \bullet T_1^{(1,2)} \bullet \ldots \bullet T_1^{(N,d)}) \bullet \ldots \bullet (T_n^{(1,1)} \bullet T_n^{(1,2)} \bullet \ldots \bullet T_n^{(N,d)}),$$

where the map $T_k^{(i,j)} : x \to y$ in each split step is

$$\begin{cases} y^{(l)} = x^{(l)}, l \neq j, \\ y^{(j)} = x^{(j)} + \Delta t v_{ij}(x, k\Delta t). \end{cases} \tag{28}$$

Here, the superscript in $x^{(l)}$ indicates the $l$-th coordinate of $x$. The map $T_k^{i,j}$ is given by the forward Euler discretization of $x'(t) = v_{i,j}(x(t), t)e_j$ in the interval $(k\Delta t, (k+1)\Delta t)$. Note that $v_{ij}$ is Lipschitz continuous on $\mathbb{R}^d$, hence the map $T_k^{i,j}$ also is.

Below is the formal statement of the approximation in this step.

**Theorem 4.1.** *Let $\Psi \in Diff_0(\Omega)$ be an orientation preserving diffeomorphism, $\Omega$ be a compact domain $\Omega \subset \mathbb{R}^d$. Then, for any $\varepsilon > 0$, there is a sequence of transformations, $T_k^{(i,j)}$, is of the form Eq. (28) such that*

$$\|\Psi(x) - (T_1^{(1,1)} \bullet T_1^{(1,2)} \bullet \ldots \bullet T_1^{(N,d)} \bullet \ldots \bullet T_n^{(1,1)} \bullet T_n^{(1,2)} \bullet \ldots \bullet T_n^{(N,d)})(x)\| \leq \varepsilon, \quad \forall x \in \Omega.$$

### 4.2 Approximate each composition component by flow maps in $H_1$ and $H_2$

Now we examine the map $T_k^{(i,j)}$ in each splitting step. Since all $T_k^{(i,j)}$ have the same structure (over a permutation), we only need to consider the case of $T_k^{(N,d)}$, which is simply denoted as $T : x \to y$ is of the form

$$T : \begin{cases} y^{(i)} = x^{(i)}, i = 1, \cdots, d-1, \\ y^{(d)} = x^{(d)} + a\sigma(w_1 x^{(1)} + \cdots + w_d x^{(d)} + b). \end{cases} \tag{29}$$

where $\sigma = \sigma_\alpha, \alpha \in (0, 1)$, is the leaky-ReLU funciton, $a, b, w_1, ..., w_d \in \mathbb{R}$ are parameters. Since the time step $\Delta t$ in $T_k^{(i,j)}$ are small, we can assume the parameters satisfing $\max(1/\alpha, \alpha)|aw_d| < 1$.

**Lemma 4.2.** *Let $\alpha > 0$ and $\max(1/\alpha, \alpha)|aw_d| < 1$, then the map $T$ in Eq. (29) is a composition of at most six flow maps in $H_1 \cup H_2$.*

Noting that the case of $w_1 = ... = w_{d-1} = 0$ is trivial, we can assume $w_1 \neq 0$ without loss of generality. Then, the bias parameter $b$ can be absorbed in $x^{(1)}$ using an affine flow map; hence we only need to consider the case of $b = 0$. In addition, using the property of leaky-ReLU, $\sigma_\alpha(x) = -\alpha\sigma_{1/\alpha}(-x)$, we can further assume $w_1 > 0$. As a result, the map $T$ can be represented by the following composition,

$$T(x) = F_0 \bullet F_1 \bullet \cdots \bullet F_5(x), \tag{30}$$

where each composition step is as follows,

$$
\begin{pmatrix} x^{(1)} \\ x^{(2:d-1)} \\ x^{(d)} \end{pmatrix} \xrightarrow{F_0} \begin{pmatrix} \nu \\ x^{(2:d-1)} \\ x^{(d)} \end{pmatrix} \xrightarrow{F_1} \begin{pmatrix} \sigma(\nu) \\ x^{(2:d-1)} \\ x^{(d)} \end{pmatrix} \xrightarrow{F_2} \begin{pmatrix} \sigma(\nu) \\ x^{(2:d-1)} \\ x^{(d)} + a\sigma(\nu) \end{pmatrix} \xrightarrow{F_3} \begin{pmatrix} \nu \\ x^{(2:d-1)} \\ x^{(d)} + a\sigma(\nu) \end{pmatrix}
$$
$$
\xrightarrow{F_4} \begin{pmatrix} \nu + w_d a\sigma(\nu) \\ x^{(2:d-1)} \\ x^{(d)} + a\sigma(\nu) \end{pmatrix} \xrightarrow{F_5} \begin{pmatrix} x^{(1)} \\ x^{(2:d-1)} \\ x^{(d)} + a\sigma(\nu) \end{pmatrix}.
$$

Here, $\nu := w_1 x^{(1)} + \cdots + w_d x^{(d)}$ and $x^{(2:d-1)}$ represent the elements $x^{(2)}, ..., x^{(d-1)}$.

We clarify that each component $F_i, i = 0, \cdots, 5$, are flow maps in $H_1 \cup H_2$. In fact, $F_0, F_2, F_5 = F_0^{-1}$ are affine mappings,

$$
F_0(x) = \begin{pmatrix} w_1 & w_{2:d} \\ 0 & I_{d-1} \end{pmatrix} x, \quad F_2(x) = \begin{pmatrix} I_{d-1} & 0 \\ (a, 0_{2:d-1}) & 1 \end{pmatrix} x, \quad F_5(x) = \begin{pmatrix} 1/w_1 & -w_{2:d}/w_1 \\ 0 & I_{d-1} \end{pmatrix} x,
$$

where $I_{d-1}$ is the identity matrix, $(a, 0_{2:d-1}) = (a, 0, ..., 0)$ with $d-2$ zeros. According to Proposition 3.1, they are flow maps in $H_1$. In addition, according to Proposition 3.2, $F_1, F_3$ and $F_4$ are leaky-ReLU flow maps in $H_2$ as

$$
F_1 = \Sigma_{(\alpha, 1_{2:d}), 1_{1:d}}, \quad F_3 = \Sigma_{(1/\alpha, 1_{2:d}), 1_{1:d}}, \quad F_4 = \Sigma_{(1+w_d a\alpha, 1_{2:d}), (1+w_d a, 1_{2:d})}. \tag{31}
$$

Here, the condition $\max(1/\alpha, \alpha)|aw_d| < 1$ is used to ensure $1 + w_d a\alpha > 0$ and $1 + w_d a > 0$.

### 4.3 Finish the construction

Combining Theorem 4.1 and Lemma 4.2 above, and using the fact in Lemma 2.7, we have the following result.

**Theorem 4.3.** *Let $\Psi \in Diff_0(\Omega)$ be an orientation preserving diffeomorphism, $\Omega$ be a compact domain $\Omega \subset \mathbb{R}^d$. Then, for any $\varepsilon > 0$, there is a sequence of flow maps, $h_1, h_2, ..., h_m, m \in \mathbb{Z}_+$, in $H = H_1 \cup H_2$ such that*

$$
\|\Psi(x) - (h_1 \bullet h_2 \bullet ... \bullet h_m)(x)\| \le \varepsilon, \quad \forall x \in \Omega. \tag{32}
$$

Then we can finish the construction for Theorem 2.6 by combining Corollary 3.7 and Theorem 4.3.

## 5 Conclusion

Universal approximation properties are the foundation for machine learning. This paper examined the approximation property of mapping composition from a sequential perspective. We proved, for the first time, that the universal approximation for diffeomorphisms and high-dimensional continuous functions can be achieved by using a finite number of sequential mappings. Our result implies that the universal approximations can be easily achieved. Importantly, the mappings used in our composition are flow maps of dynamical systems and do not increase the dimensions. However, our result is restricted to mappings on a compact domain. It is interesting to study whether it is possible to generalize this result to the case of mappings on unbounded domains.

Our Theorem 2.2 was inspired by the fact of finite vocabulary in natural languages, where $V$ can be mimicked to a "vocabulary", $H_1$ and $H_2$ to "phrases", and $\mathcal{H}_V$ to "sentences". Our results provide a novel aspect for composite mappings, and we hope our findings could in turn inspire related research for the algorithm and modeling communities. For example, one can embed words as nonlinear mappings instead of vectors or matrices in traditional models. However, constructing such embedding models involves lots of techniques that are beyond the scope of this paper.

It should be noted that this paper focuses on the existence of a finite vocabulary and the constructed $V$ in Eq. (8) is not optimal. If a sequential composition of mappings in such $V$ is used to approximate functions in practical applications, the required sequence length may be extremely large. However, in practical applications, it is often only necessary to approximate a certain small set of continuous functions, hence designing an efficient vocabulary for them would be a fascinating future direction.

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

# A    ADDITIONAL LEMMAS

It is well known that an ODE system can be approximated by many numerical methods. Particularly, we use the splitting approach (Holden et al., 2010). Let $v(x,t)$ be the summation of several functions,

$$v(x,t) = \sum_{j=1}^{J} v_j(x,t), J \in \mathbb{Z}_+. \tag{33}$$

For a given time step $\Delta t$, we define the iteration as

$$x_{k+1} = T_k x_k = T_k^{(J)} \circ \cdots \circ T_k^{(2)} \circ T_k^{(1)} x_k, \tag{34}$$

where the map $T_k^{(j)} : x \to y$ is

$$y \equiv T_k^{(j)}(x) = x + \Delta t v_j(x, t_k) \quad t_k = k\Delta t. \tag{35}$$

**Lemma A.1.** *Let all $v_j(x,t), j = 1, 2, ..., J$, and $v(x,t), (x,t) \in \mathbb{R}^d \times [0, \tau]$, be piecewise constant (w.r.t. $t$) and $L$-Lipschitz ($L > 0$). Then, for any $\tau > 0$, $\varepsilon > 0$ and $x_0$ in a compact domain $\Omega$, there exist a positive integer $n$ and $\Delta t = \tau/n < 1$ such that $\|x(\tau) - x_n\| \le \varepsilon$.*

*Proof.* Without loss of generality, we only consider $J = 2$. (The general $J$ case can be proven accordingly.) In addition, we assume $v_j$ are constant w.r.t. $t$ in each interval $[t_k, t_{k+1})$ the time step, *i.e.* $v(x,t) = v(x, t_k), t \in [t_k, t_{k+1})$. This can be arrived at by choosing small enough $\Delta t$ and adjusting the time step to match the piecewise points of $v_j$. Thus, we have

$$\begin{aligned} x_{k+1} &= T_k^{(2)}(x_k + \Delta t v_1(x_k, t_k)) \\ &= x_k + \Delta t v_1(x_k, t_k) + \Delta t v_2(x_k + \Delta t v_1(x_k, t_k), t_k) \\ &= x_k + \Delta t(v_1(x_k, t_k) + v_2(x_k, t_k)) + \Delta t^2 R_k. \end{aligned}$$

Since $v_i(x,t)$ are Lipschitz, the residual term $R_k$ is bounded by a constant $R$ that is independent of $k$. In fact, we have

$$\begin{aligned} \|R_k\| &= \|v_2(x_k + \Delta t v_1(x_k, t_k), t_k) - v_2(x_k, t_k)\|/\Delta t \\ &\le L\|v_1(x_k, t_k)\| \le L(\|v_1(0, t_k)\| + L\|x_k\|). \end{aligned}$$

Let $V := \sup\{\|v_j\| | t \in (0, \tau)\}, X := \sup\{\|x_0\| | x_0 \in \Omega\}$, then we have

$$\|x_{k+1}\| \le (1 + L\Delta t)\|x_k\| + \Delta t^2 L(V + L\|x_k\|). \tag{36}$$

As a result, $\|x_k\|$ is bounded by $B := (X + \frac{V\Delta t}{1+L})e^{L(1+L)\tau}$ and $\|R_k\|$ is bounded by $R := L(V + LB)$.

Using the integral form of the ODE and defining the error as $e_k := x_k - x(t_k)$, we have the following estimation:

$$\begin{aligned} \|e_{k+1}\| &= \|e_k + \int_{t_k}^{t_{k+1}} (v(x_k, t_k) - v(x(t), t))dt + R_k \Delta t^2\| \\ &\le \|e_k\| + \int_{t_k}^{t_{k+1}} \|v(x(t), t_k) - v(x_k, t_k)\|dt + \|R_k\|\Delta t^2 \\ &\le (1 + L\Delta t)\|e_k\| + R\Delta t^2. \end{aligned}$$

Employing the inequality $(1 + L\Delta t)^k \le e^{Lk\Delta t} \le e^{L\tau}$ and the initial error $e_0 = 0$, we have

$$\|e_k\| \le (1 + L\Delta t)^k \|e_0\| + \frac{R\Delta t^2}{L\Delta t}[(1 + L\Delta t)^k - 1] \le R\Delta t(e^{L\tau} - 1)/L. \tag{37}$$

For any $\varepsilon > 0$, let $n \ge \lceil \frac{R\tau e^{L\tau}}{L\varepsilon} \rceil$, then we have $\|x(t_k) - x_k\| \le \varepsilon$, which finishes the proof. $\quad\square$

# B PROOFS OF LEMMAS AND PROPOSITIONS

## B.1 PROOF OF LEMMA 2.7

**Lemma 2.7.** *Let map $T = F_1 \bullet ... \bullet F_n$ be a composition of $n$ continuous functions $F_i$ defined on an open domain $D_i$, and let $\mathcal{F}$ be a continuous function class that can uniformly approximate each $F_i$ on any compact domain $\mathcal{K}_i \subset D_i$. Then, for any compact domain $\mathcal{K} \subset D_1$ and $\varepsilon > 0$, there are $n$ functions $\tilde{F}_1, ..., \tilde{F}_n$ in $\mathcal{F}$ such that*

$$\|T(x) - \tilde{F}_1 \bullet ... \bullet \tilde{F}_n(x)\| \le \varepsilon, \quad \forall x \in \mathcal{K}. \tag{38}$$

*Proof.* It is enough to prove the case of $n = 2$. (The case of $n > 2$ can be proven by the method of induction, as $T$ can be expressed as the composition of two functions, $T = F_n \circ T_{n-1}$, with $T_{n-1} = F_{n-1} \circ ... \circ F_1$.) According to the definition, we have $F_1(D_1) \subset D_2$. Since $D_2$ is open and $F_1(\mathcal{K})$ is compact, we can choose a compact set $\mathcal{K}_2 \subset D_2$ such that $\mathcal{K}_2 \supset \{F_1(x) + \delta_0 y : x \in \mathcal{K}, \|y\| < 1\}$ for some $\delta_0 > 0$ that is sufficiently small.

According to the continuity of $F_2$, there is a $\delta \in (0, \delta_0)$ such that

$$\|F_2(y) - F_2(y')\| \le \varepsilon/2, \forall y, y' \in \mathcal{K}_2,$$

provided $\|y - y'\| \le \delta$. The approximation property of $\mathcal{F}$ allows us to choose $\tilde{F}_1, \tilde{F}_2 \in \mathcal{F}$ such that

$$\|\tilde{F}_1(x) - F_1(x)\| \le \delta < \delta_0, \quad \forall x \in \mathcal{K},$$
$$\|\tilde{F}_2(y) - F_2(y)\| \le \varepsilon/2, \quad \forall y \in \mathcal{K}_2.$$

As a consequence, for any $x \in \mathcal{K}$, we have $F_1(x), \tilde{F}_1(x) \in \mathcal{K}_2$ and

$$\|F_2 \circ F_1(x) - \tilde{F}_2 \circ \tilde{F}_1(x)\| \le \|F_2 \circ F_1(x) - F_2 \circ \tilde{F}_1(x)\| + \|F_2 \circ \tilde{F}_1(x) - \tilde{F}_2 \circ \tilde{F}_1(x)\|$$
$$\le \varepsilon/2 + \varepsilon/2 = \varepsilon.$$

$\square$

## B.2 PROOF OF PROPOSITION 3.1

**Proposition 3.1.** *(1) Let $Q$ be a nonsingular matrix. If $x \to Wx$ is an affine flow map then the map $x \to QWQ^{-1}x$, $x \to W^T x$ and $x \to W^{-1}x$ also are. (2) Let $U$ be an upper triangular matrix below with $\lambda > 0$, then the map $x \to Ux$ is an affine flow map for arbitrary vector $w_{2:d}$,*

$$U = \begin{pmatrix} \lambda & w_{2:d} \\ 0 & I_{d-1} \end{pmatrix}. \tag{39}$$

*Proof.* (1) It is because $\ln(QWQ^{-1}) = Q\ln(W)Q^{-1}$, $\ln(W^T) = \ln(W)^T$ and $\ln(W^{-1}) = -\ln(W)$ are real as $\ln(W)$ is real. (2) It can be obtained by employing the formula,

$$\ln \begin{pmatrix} \lambda & w_{2:d} \\ 0 & I_{d-1} \end{pmatrix} = \begin{pmatrix} \ln(\lambda) & \frac{\ln(\lambda)}{\lambda-1} w_{2:d} \\ 0 & 0 \end{pmatrix}, \quad \lambda \ne 1. \tag{40}$$

When $\lambda = 1$, the formula is simplified as $\ln(U) = U - I_d$.

$\square$

## B.3 PROOF OF PROPOSITION 3.2

**Proposition 3.2.** *If $\tilde{\alpha}, \tilde{\beta} \in (0, \infty)^d$, then the map $\Sigma_{\tilde{\alpha}, \tilde{\beta}}$ is a leaky-ReLU flow map.*

*Proof.* By directly calculate the flow map $\phi^\tau_{\Sigma_{\alpha,\beta}(x)}$ with $\alpha, \beta \in \mathbb{R}^d$, we have

$$\phi^\tau_{\Sigma_{\alpha,\beta}(x)}(x) = \Sigma_{\tilde{\alpha}, \tilde{\beta}}(x), \tag{41}$$

where $\tilde{\alpha} = (e^{\tau\alpha_1}, ..., e^{\tau\alpha_d})$ and $\tilde{\beta} = (e^{\tau\beta_1}, ..., e^{\tau\beta_d})$. Choosing $\alpha_i = \ln(\tilde{\alpha}_i), \beta_i = \ln(\tilde{\beta}_i)$ and $\tau = 1$, we can finish the proof. $\square$

### B.4 PROOF OF LEMMA 3.4

**Theorem 3.3. (Lie product formula)** *For all matrix $A, B \in \mathbb{R}^{d \times d}$, we have*

$$e^{A+B} = \lim_{n \to \infty} \left( e^{A/n} e^{B/n} \right)^n = \lim_{n \to \infty} \left( \phi_{Ax}^{1/n} \bullet \phi_{Bx}^{1/n} \right)^{\bullet n} \tag{42}$$

*Proof.* The proof can be found in Hall (2015). □

**Lemma 3.4.** *Let $v_i : \mathbb{R}^d \to \mathbb{R}^d, i = 1, 2, ..., m$ be Lipschitz continuous funcitons, $v = \sum_{i=1}^m v_i$, $\Omega$ be a compact domain. For any $t > 0$ and $\varepsilon > 0$, there is a positive integers $n$, such that the flow map $\phi_v^t$ can be approximated by composition of flow maps $\phi_{v_i}^{t/n}$, i.e.*

$$\|\phi_v^t(x) - \left( \phi_{v_1}^{t/n} \bullet \phi_{v_2}^{t/n} \bullet ... \bullet \phi_{v_m}^{t/n} \right)^{\bullet n}(x)\| < \varepsilon, \quad \forall x \in \Omega. \tag{43}$$

*Proof.* It's a special case of Lemma A.1 with a velocity field $v_i$ independent on $t$. □

### B.5 PROOF OF LEMMA 3.6

**Theorem 3.5. (Kronecker's approximation theorem)** *Let $\gamma \in \mathbb{R}$ be an irrational number, then for any $t \in \mathbb{R}$ and $\varepsilon > 0$, there exist two integers $p$ and $q$ with $q > 0$, such that $|\gamma q + p - t| < \varepsilon$.*

*Proof.* The proof can be found in Apostol (1990). □

**Lemma 3.6.** *Let $v : \mathbb{R}^d \to \mathbb{R}^d$ be a Lipschitz continuous function, $\Omega$ be a compact domain. For any $t > 0$ and $\varepsilon > 0$, there exist two positive integers $p$ and $q$, such that the flow map $\phi_v^t$ can be approximated by $(\phi_v^1)^{\bullet p} \bullet (\phi_{-v}^{\sqrt{2}})^{\bullet q}$, i.e.*

$$\|\phi_v^t(x) - (\phi_v^1)^{\bullet p} \bullet (\phi_{-v}^{\sqrt{2}})^{\bullet q}(x)\| < \varepsilon, \quad \forall x \in \Omega. \tag{44}$$

*Proof.* Since the field $v$ is Lipschitz and the domain $\Omega$ is compact, there exist a constant $C > 0$ such that

$$\|\phi_v^{t_2}(x_0) - \phi_v^{t_1}(x_0)\| \leq \int_{t_1}^{t_2} \|v(x(t))\| dt < C|t_2 - t_1|, \quad \forall x_0 \in \Omega. \tag{45}$$

Employing the Kronecker's Theorem 3.5 with $\gamma = -\sqrt{2}$, approximating $t$ by $p - q\sqrt{2}$ such that

$$|p - q\sqrt{2} - t| < \varepsilon/C, \tag{46}$$

then we have

$$\|\phi_v^t(x) - \phi_v^{p-q\sqrt{2}}(x)\| < \varepsilon, \quad \forall x \in \Omega. \tag{47}$$

As $t$ is positive, we have $p$ is positive as $q$ is. The following representation of the flow maps finishes the proof,

$$\phi_v^{p-q\sqrt{2}} = \phi_v^p \bullet \phi_v^{-q\sqrt{2}} = \phi_v^p \bullet \phi_{-v}^{q\sqrt{2}} = (\phi_v^1)^{\bullet p} \bullet (\phi_{-v}^{\sqrt{2}})^{\bullet q}. \tag{48}$$

□

**Corollary 3.7.** *For any flow maps $h$ in $H_1 \cup H_2$, $\varepsilon > 0$ and compact domain $\Omega \subset \mathbb{R}^d$, there is a sequence $\phi_1, \phi_2, ..., \phi_m$ in $V$ (Eq. 8) such that*

$$\|h(x) - (\phi_1 \bullet \phi_2 \ldots \bullet \phi_m)(x)\| < \varepsilon, \quad \forall x \in \Omega. \tag{49}$$

*Proof.* The proof is finished by directly employing Lemma 3.4 and Lemma 3.6 with the following splittings,

$$Ax + b = \sum_{i=1}^d \sum_{j=1}^d a_{ij} E_{ij} x + \sum_{i=1}^d b_i e_i, \quad \Sigma_{\boldsymbol{\alpha},\boldsymbol{\beta}}(x) = \sum_{i=1}^d \alpha_i \Sigma_{e_i,0}(x) + \sum_{i=1}^d \beta_i \Sigma_{0,e_i}(x). \tag{50}$$

□

### B.6 PROOF OF LEMMA 4.2

**Lemma 4.2.** *Let $\alpha > 0$ and $\max(1/\alpha, \alpha)|aw_d| < 1$, then the map $T$ in Eq. (29) is a composition of at most six flow maps in $H_1 \cup H_2$.*

*Proof.* Recall the map $T : x \to y$ is of the form

$$T : \begin{cases} y^{(i)} = x^{(i)}, i = 1, \cdots, d-1, \\ y^{(d)} = x^{(d)} + a\sigma(w_1 x^{(1)} + \cdots + w_d x^{(d)} + b). \end{cases} \tag{51}$$

where $\sigma = \sigma_\alpha$ is the leaky-ReLU funciton, $a, b, w_1, ..., w_d \in \mathbb{R}$ are parameters. We construct the composition flow maps in three cases.

(1) The case of $w_1 = ... = w_d = 0$. In this case, $T$ is already an affine flow map in $H_1$.

(2) The case of $w_1 = ... = w_{d-1} = 0, w_d \neq 0$. In this case, we only need to consider the last coordinate as the first $d-1$ coordinates are kept. According to

$$y^{(d)} = x^{(d)} + a\sigma_\alpha(w_d x^{(d)} + b) = (x^{(d)} + \tfrac{b}{w_d}) + a\sigma_\alpha(w_d(x^{(d)} + \tfrac{b}{w_d})) - \tfrac{b}{w_d}, \tag{52}$$

we can assume $b = 0$ as it can be absorbed in an affine flow map. Let $\tilde{\alpha} = 1 + \alpha a w_d > 0, \tilde{\beta} = 1 + a w_d > 0$, as $\max(1/\alpha, \alpha)|aw_d| < 1$, we have the following representation,

$$x^{(d)} + a\sigma_\alpha(w_d x^{(d)}) = \begin{cases} \sigma_{\tilde{\alpha}, \tilde{\beta}}(x^{(d)}), & w_d < 0, \\ \sigma_{\tilde{\beta}, \tilde{\alpha}}(x^{(d)}), & w_d > 0, \end{cases} \tag{53}$$

which is a leaky-ReLU flow map in $H_3$ either $w_d > 0$ or $w_d < 0$.

(3) The case of $w_i \neq 0$ for some $i = 1, ..., d-1$. We only show the case of $w_1 \neq 0$ without loss of generality. Same with (1), we can absorb $b$ in $x^{(1)}$ using an affine flow map; hence we only need to consider the case of $b = 0$. In addition, using the property of leaky-ReLU,

$$\sigma_\alpha(x) = -\alpha\sigma_{1/\alpha}(-x) \tag{54}$$

$\sigma_\alpha(x) = -\alpha\sigma_{1/\alpha}(-x)$, we can further assume $w_1 > 0$. (If $w_1 < 0$, we change $w$ to $-w$, $\alpha$ to $1/\alpha$, $a$ to $a\alpha$, which does not change the map $T$). As a result, the map $T$ can be represented by the following composition,

$$T(x) = F_0 \bullet F_1 \bullet \cdots \bullet F_5(x), \tag{55}$$

where each composition step is as follows,

$$\begin{pmatrix} x^{(1)} \\ x^{(2:d-1)} \\ x^{(d)} \end{pmatrix} \xrightarrow{F_0} \begin{pmatrix} \nu \\ x^{(2:d-1)} \\ x^{(d)} \end{pmatrix} \xrightarrow{F_1} \begin{pmatrix} \sigma(\nu) \\ x^{(2:d-1)} \\ x^{(d)} \end{pmatrix} \xrightarrow{F_2} \begin{pmatrix} \sigma(\nu) \\ x^{(2:d-1)} \\ x^{(d)} + a\sigma(\nu) \end{pmatrix} \xrightarrow{F_3} \begin{pmatrix} \nu \\ x^{(2:d-1)} \\ x^{(d)} + a\sigma(\nu) \end{pmatrix}$$

$$\xrightarrow{F_4} \begin{pmatrix} \nu + w_d a\sigma(\nu) \\ x^{(2:d-1)} \\ x^{(d)} + a\sigma(\nu) \end{pmatrix} \xrightarrow{F_5} \begin{pmatrix} x^{(1)} \\ x^{(2:d-1)} \\ x^{(d)} + a\sigma(\nu) \end{pmatrix}.$$

Here, $\nu := w_1 x^{(1)} + \cdots + w_d x^{(d)}$ and $x^{(2:d-1)}$ represent the elements $x^{(2)}, ..., x^{(d-1)}$. We clarify that each component $F_i, i = 0, \cdots, 5$, are flow maps in $H_1 \cup H_2$.

In fact, $F_0, F_2, F_5 = F_0^{-1}$ are affine transformations,

$$F_0(x) = \begin{pmatrix} w_1 & w_{2:d} \\ 0 & I_{d-1} \end{pmatrix} x, \quad F_2(x) = \begin{pmatrix} I_{d-1} & 0 \\ (a, 0_{2:d-1}) & 1 \end{pmatrix} x, \quad F_5(x) = \begin{pmatrix} 1/w_1 & -w_{2:d}/w_1 \\ 0 & I_{d-1} \end{pmatrix} x,$$

where $I_{d-1}$ is the identity matrix, $(a, 0_{2:d-1}) = (a, 0, ..., 0)$ with $d-2$ zeros. According to Proposition 3.1, they are flow maps in $H_1$. In addition, $F_1, F_3$ and $F_4$ are leaky-ReLU flow maps in $H_2$ as

$$F_1 = \Sigma_{(\alpha, 1_{2:d}), 1_{1:d}}, \quad F_3 = \Sigma_{(1/\alpha, 1_{2:d}), 1_{1:d}}, \quad F_4 = \Sigma_{(1+w_d a\alpha, 1_{2:d}), (1+w_d a, 1_{2:d})}. \tag{56}$$

Here, the condition $\max(1/\alpha, \alpha)|aw_d| < 1$ is used to ensure $1 + w_d a\alpha > 0$ and $1 + w_d a > 0$, no matter whether Eq. (54) is uesd.

$\square$

## C  PROOF OF THE MAIN THEOREMS

### C.1  PROOF OF THEOREM 4.1

**Theorem 4.1.** *Let* $\Psi \in Diff_0(\Omega)$ *be an orientation preserving diffeomorphism,* $\Omega$ *be a compact domain* $\Omega \subset \mathbb{R}^d$. *Then, for any* $\varepsilon > 0$, *there is a sequence of transformations,* $T_k^{(i,j)}$, *is of the form Eq. (28) such that*

$$\|\Psi(x) - (T_1^{(1,1)} \bullet T_1^{(1,2)} \bullet \ldots \bullet T_1^{(N,d)} \bullet \ldots \bullet T_n^{(1,1)} \bullet T_n^{(1,2)} \bullet \ldots \bullet T_n^{(N,d)})(x)\| \leq \varepsilon, \quad \forall x \in \Omega.$$

*Proof.* (1) Firstly, employed results of Agrachev & Caponigro (2010) and Caponigro (2011), any OP diffeomorphism $\Psi$ can be approximated by flow map of ODEs. Particularly, we can choose the ODEs as neural ODEs are of the form

$$x'(t) = v(x(t), t) = \sum_{i=1}^{N} s_i(t)\sigma(w_i(t) \cdot x(t) + b_i(t)), \tag{57}$$

where the field function $v$ is a neural network with $N$ hidden neurons, the activation is chosen as the leaky-ReLU function $\sigma = \sigma_\alpha$ for some $\alpha \in (0,1)$, $s_i \in \mathbb{R}^d$, $w_i \in \mathbb{R}^d$ and $b_i \in \mathbb{R}$ are piecewise constant functions of $t$. The universal approximation property of neural networks Cybenko (1989) implies that, for any $\varepsilon > 0$, there exist $s_i \in \mathbb{R}^d$, $w_i \in \mathbb{R}^d$, $b_i \in \mathbb{R}, \tau > 0$ and $N \in \mathbb{Z}_+$, such that

$$\|\Psi(x) - \phi_v^\tau(x)\| < \varepsilon/2, \quad \forall x \in \Omega, \tag{58}$$

where $\phi_v^\tau$ is the flow map of Eq. (27).

(2) Following the approach of Duan et al. (2022), we employ a proper splitting numerical scheme to discretize the neural ODE (27). Split the field $v$ as a summation of $Nd$ functions, $v(x,t) = \sum_{i=1}^{N} \sum_{j=1}^{d} v_{ij}(x,t)e_j$, where $e_j$ is the $j$-th axis unit vector and $v_{ij}(x,t) = s_{ij}(t)\sigma(w_i(t) \cdot x + b_i(t))$ are scalar Lipschitz functions. Then Lemma A.1 implies that there is a $n \in \mathbb{Z}_+$ big enough such that

$$\|\phi_v^\tau(x) - \Phi(x)\| < \varepsilon/2, \quad \forall x \in \Omega, \tag{59}$$

where

$$\Phi = T_1 \bullet \cdots \bullet T_n \equiv (T_1^{(1,1)} \bullet T_1^{(1,2)} \bullet \ldots \bullet T_1^{(N,d)}) \bullet \ldots \bullet (T_n^{(1,1)} \bullet T_n^{(1,2)} \bullet \ldots \bullet T_n^{(N,d)}),$$

and the map $T_k^{(i,j)} : x \to y$ is of the form

$$\begin{cases} y^{(l)} = x^{(l)}, l \neq j, \\ y^{(j)} = x^{(j)} + \Delta t v_{ij}(x, k\Delta t). \end{cases} \tag{60}$$

Here, the superscript in $x^{(l)}$ indicates the $l$-th coordinate of $x$.

(3) Combining the above two parts, we finish the proof. $\square$

### C.2  PROOF OF THEOREM 4.3

**Theorem 4.3.** *Let* $\Psi \in Diff_0(\Omega)$ *be an orientation preserving diffeomorphism,* $\Omega$ *be a compact domain* $\Omega \subset \mathbb{R}^d$. *Then, for any* $\varepsilon > 0$, *there is a sequence of flow maps,* $h_1, h_2, ..., h_m, m \in \mathbb{Z}_+$, *in* $H = H_1 \cup H_2$ *such that*

$$\|\Psi(x) - (h_1 \bullet h_2 \bullet ... \bullet h_m)(x)\| \leq \varepsilon, \quad \forall x \in \Omega. \tag{61}$$

*Proof.* According to Theorem 4.1, there is a sequence of transformations, $T_k^{(i,j)}$, is of the form Eq. (28) such that

$$\|\Psi(x) - (T_1^{(1,1)} \bullet T_1^{(1,2)} \bullet \ldots \bullet T_1^{(N,d)} \bullet \ldots \bullet T_n^{(1,1)} \bullet T_n^{(1,2)} \bullet \ldots \bullet T_n^{(N,d)})(x)\| \leq \varepsilon, \quad \forall x \in \Omega.$$

Here $n$ can be choosed large enough such that $\max(1/\alpha, \alpha)C^2 \Delta t < 1, \Delta t = \tau/n$, where

$$C = \max_{t \in [0,\tau]} \{|s_{ij}(t)|, |w_{ij}(t)| \mid i, j = 1, 2, ..., d\}. \tag{62}$$

Since $s_i, w_i$ are piecewise constant functions, the constant $C$ is finite. Then according to Lemma 4.2, each $T_k^{(i,j)}$ is a composition of at most six flow maps in $H_1 \cup H_2$. As a consequence, we finish the proof by relabelling the index of the used flow maps.

$\square$

## C.3 PROOF OF THEOREM 2.6

**Theorem 2.6.** *Let $\Psi \in Diff_0(\Omega)$ be an orientation preserving diffeomorphism, $\Omega$ be a compact domain $\Omega \subset \mathbb{R}^d$. Then, for any $\varepsilon > 0$, there is a sequence of flow maps, $\phi_1, \phi_2, ..., \phi_n \in V, n \in \mathbb{Z}_+$, such that*

$$\|\Psi(x) - (\phi_1 \bullet \phi_2 \bullet ... \bullet \phi_n)(x)\| \leq \varepsilon, \quad \forall x \in \Omega. \tag{63}$$

*Proof.* (1) According to Theorem 4.3, there is a sequence of flow maps, $h_1, h_2, ..., h_m, m \in \mathbb{Z}_+$, in $H = H_1 \cup H_2$ such that

$$\|\Psi(x) - (h_1 \bullet h_2 \bullet ... \bullet h_m)(x)\| \leq \varepsilon/2, \quad \forall x \in \Omega. \tag{64}$$

(2) According to Corollary 3.7, each $h_i$ can be universal approximation by $\mathcal{H}_V$, *i.e.*, for any $\varepsilon_i > 0$ and compact domain $\Omega_i$, there is a sequence of flow maps, $\phi_{i,1}, ..., \phi_{i,n_i} \in V$, such that

$$\|h_i(x) - (\phi_{i,1} \bullet \phi_{i,2} \bullet ... \bullet \phi_{i,n_i})(x)\| \leq \varepsilon_i, \quad \forall x \in \Omega_i. \tag{65}$$

(3) According to Lemma 2.7, we can choose $\phi_{i,j} \in V$ and reindex them as $\phi_1, \phi_2, ..., .\phi_n$ such that

$$\|(h_1 \bullet h_2 \bullet ... \bullet h_m)(x) - (\phi_1 \bullet \phi_2 \bullet ... \bullet \phi_n)(x)\| \leq \varepsilon/2, \quad \forall x \in \Omega. \tag{66}$$

(4) Combining (1) and (3), we finish the proof.

$\square$

## C.4 PROOF OF THEOREM 2.2

**Theorem 2.2.** *Let $\Omega \subset \mathbb{R}^d$ be a compact domain. Then, there is a finite set $V \subset Diff_0(\mathbb{R}^d)$ such that the hypothesis space $\mathcal{H}_V$ in Eq. (3) has C-UAP for $Diff_0(\mathbb{R}^d)$.*

*Proof.* The Theorem 2.6 provides constructive proof for the existence of $V$ in Eq. (8). $\square$

**Corollary 2.3.** *Let $\Omega \subset \mathbb{R}^d$ be a compact domain, $d \geq 2$ and $p \in [1, +\infty)$. Then, there is a finite set $V \subset C(\mathbb{R}^d, \mathbb{R}^d)$ such that the hypothesis space $\mathcal{H}_V$ in Eq. (3) has $L^p$-UAP for $C(\mathbb{R}^d, \mathbb{R}^d)$.*

*Proof.* We can use the same $V$ in Theorem 2.2 as $V \subset \text{Diff}_0(\mathbb{R}^d) \subset C(\mathbb{R}^d, \mathbb{R}^d)$.

(1) Let $f \in C(\mathbb{R}^d, \mathbb{R}^d), d \geq 2$, then the result of Brenier & Gangbo (2003) indicates that for any $\varepsilon > 0$, there is a OP diffeomorphism $\Psi \in \text{Diff}_0(\mathbb{R}^d)$ such that

$$\|f - \Psi\|_{L^p(\Omega)} < \varepsilon/2. \tag{67}$$

(2) The Theorem 2.2 indicates that, there is mapping $\Phi \in \mathcal{H}_V$ such that

$$\|\Psi(x) - \Phi(x)\| < \varepsilon' = \frac{\varepsilon}{2|\Omega|}, \quad \forall x \in \Omega. \tag{68}$$

(3) Combining (1) and (2), we have

$$\|f - \Phi\|_{L^p(\Omega)} < \varepsilon. \tag{69}$$

which finishes the proof. $\square$

# D  VOCABULARY FOR LINEAR SPACES

Here we provide similar results for both the vector space and the linear mapping space. Note that linear mappings can be characterized as matrics and the construction here is much simpler than what we do in the main body of this paper for the continuous function space.

**Theorem D.1.** *There is a finite set $V_0 \subset \mathbb{R}^d$, such that for any vector $v^* \in \mathbb{R}^d$ and $\varepsilon > 0$, there is a sequence, $v_{i_1}, v_{i_2}, ..., v_{i_n}$, in $V_0$, $n \in \mathbb{Z}_+$, such that*

$$\|v_{i_1} + v_{i_2} + ... + v_{i_n} - v^*\| < \varepsilon.$$

*Proof.* Directly employing Kronecker's Theorem 3.5, it is easy to see the following set satisfies the requirement,

$$V_0 = \{\lambda e_i | \lambda \in \{\pm 1, \pm \sqrt{2}\}, i = 1, 2, ..., d\}, \tag{70}$$

where $e_i$ is the axis vector in the $i$-th coordinate.  □

**Lemma D.2.** *Let $V_1 = \{0, \pm 1, 10^{\pm 1}, \pm 10^{\pm \sqrt{2}}\}$, then for any number $\lambda \in \mathbb{R}$ and $\varepsilon > 0$, there is a sequence, $v_{i_1}, v_{i_2}, ..., v_{i_n}$, in $V_1$, $n \in \mathbb{Z}_+$, such that*

$$|v_{i_1} v_{i_2} ... v_{i_n} - \lambda| < \varepsilon.$$

*Proof.* It is enough to consider the case of $\lambda > 0$. According to Theorem D.1 with $d = 1$, we can finish the proof by approximating $v^* = \log_{10}(\lambda)$.  □

**Theorem D.3.** *There is a finite set $V_2 \subset \mathbb{R}^{d \times d}$, such that for any matrix $A^* \in \mathbb{R}^{d \times d}$ and $\varepsilon > 0$, there is a sequence, $A_{i_1}, A_{i_2}, ..., A_{i_n}$, in $V_2$, $n \in \mathbb{Z}_+$, such that*

$$\|A_{i_1} A_{i_2} ... A_{i_n} - A^*\| < \varepsilon.$$

*Proof.* For simplicity, we only consider the case of $d = 2$ as the general cases can be proved in the same way. Since any singular matrix can be approximated by nonsingular matrixes, we only need to consider $A^*$ as a nonsingular matrix. In addition, every nonsingular matrix can be represented as a product of elementary matrices. Hence we can further assume $A^*$ to be an elementary matrix. Note that the elementary matrices are of the following,

$$\begin{pmatrix} \lambda & 0 \\ 0 & 1 \end{pmatrix}, \begin{pmatrix} 1 & 0 \\ \lambda & 1 \end{pmatrix}, \begin{pmatrix} 0 & 1 \\ 1 & 0 \end{pmatrix}, \lambda \neq 0.$$

Therefore, we can finish the proof by considering the following set $V_2$,

$$V_2 = \left\{ \begin{pmatrix} \lambda & 0 \\ 0 & 1 \end{pmatrix}, \begin{pmatrix} 1 & 0 \\ 1 & 1 \end{pmatrix}, \begin{pmatrix} 0 & 1 \\ 1 & 0 \end{pmatrix} \middle| \lambda \in \{\pm 1, 10^{\pm 1}, 10^{\pm \sqrt{2}}\} \right\}. \tag{71}$$

The validation of this $V_2$ can be verified by using Lemma D.2 and the following relations,

$$\begin{pmatrix} 1 & 0 \\ \lambda & 1 \end{pmatrix} = \begin{pmatrix} 1/\lambda & 0 \\ 0 & 1 \end{pmatrix} \begin{pmatrix} 1 & 0 \\ 1 & 1 \end{pmatrix} \begin{pmatrix} \lambda & 0 \\ 0 & 1 \end{pmatrix}, \lambda \neq 0,$$

$$\begin{pmatrix} \lambda_1 & 0 \\ 0 & 1 \end{pmatrix} \begin{pmatrix} \lambda_2 & 0 \\ 0 & 1 \end{pmatrix} = \begin{pmatrix} \lambda_1 \lambda_2 & 0 \\ 0 & 1 \end{pmatrix}, \lambda_1, \lambda \in \mathbb{R}.$$

□

