# OpenReview forum: "Vocabulary for Universal Approximation: A Linguistic Perspective of Mapping Compositions"
_ICLR.cc/2024/Conference — Submitted to ICLR 2024_

### Official Review · Reviewer_JbK3 · 2023-10-31

**Soundness:** 3 good
**Presentation:** 2 fair
**Contribution:** 3 good
**Rating:** 6
**Confidence:** 1

**Summary:**

This paper examined the approximation property of mapping composition from a sequential perspective.
The author claim to prove the universal approximation for diffeomorphisms can be achieved by using a finite number of sequential mappings for the first time. And the results show that the universal approximations can be easily achieved.

**Strengths:**

The paper focus on an important problem: the application of sequential model to various morphism has only to empirically studied but never been theoretically proved.   So if the Theorems in this paper is right, it could be a missing piece to the puzzle.
Honestly I am not theoretically sound enough to understand this paper, so I can't give more valuable suggestion to this paper.

**Weaknesses:**

It would be the paper more convincible if the author could add some empirical experiments and analysis to show more evidence to the theory.

**Questions:**

Honestly I am not theoretically sound enough to understand this paper, so I can't give more valuable questions to this paper.

---

> ### Author Response · Authors · 2023-11-18
> **Response to Reviewer JbK3**
>
> We thank the reviewer for the kind comments and suggestions. We have revised the paper to include an additional subsection to explain relevant mathematical concepts, aiming to enhance the readability of the theoretical parts.
>
> Regarding experiments, our results suggest embedding words as functions rather than vectors. Executing such experiments is presently beyond the scope of our capabilities, and thus, we leave it as future work. Fortunately, the reviewer gE5q has highlighted that there are existing experiments utilizing mapping as word embedding, such as the Compositional Matrix Space Model (CMSM). Despite these experiments being restricted to linear mappings, they have demonstrated certain advantages, such as the ability to capture word order. CMSM can be considered as experimental evidence supporting some of the points made in this paper. Concurrently, this paper can serve as a theoretical underpinning for CMSM from the approximation aspect. We have included this point in our revised paper.

---

### Official Review · Reviewer_gE5q · 2023-11-01

**Soundness:** 4 excellent
**Presentation:** 3 good
**Contribution:** 4 excellent
**Rating:** 8
**Confidence:** 2

**Summary:**

The paper presents a constructive proof of a universal approximation result, namely that any mapping can be represented as the composition of functions selected from a finite set of functions, drawing analogies to how any sentence can be build from a finite set of words.The main idea of the constructive proof is to demonstrate that a sequence of flow maps derived from ODEs can be combined to approximate any given continuous mapping.

**Strengths:**

* Important and intuitive theoretical result
* Solid motivation through the universality of language
* Good contextualization with results from related works
* Certainly relevant to some parts of the ICLR community

I can't reasonably comment on the correctness of the proof, since I am not familiar with most of the literature or techniques used.

**Weaknesses:**

* The paper could use a "Preliminaries" section to introduce the non-expert reader to some of the relevant concepts, such as ODEs and orientation-preserving diffeomorphisms. Instead, the paper refers to a textbook introduction to dynamical systems, which is not practical. Only slightly more than 8 / 9 pages were used, so there is almost 1 page of space left for these kinds of things.

Minor:
* use \citep and \citet where appropriate

**Questions:**

CMOW is a simple sentence embedding method that represents every word as a matrix and composes words in a sentence via matrix multiplication. Hence, CMOW builds a linear deep neural network dynamically. Could you comment on in how far this relates to your idea in the conclusion to construct novel NLP models by embedding every word as a function?

CBOW Is Not All You Need: Combining CBOW with the Compositional Matrix Space Model
https://openreview.net/forum?id=H1MgjoR9tQ

---

> ### Author Response · Authors · 2023-11-18
> **Response to Reviewer gE5q**
>
> We thank the reviewer for the careful reading and the constructive comments and suggestions. Especially bringing the excellent paper on CMOW to our attention.
>
> 1. Preliminaries. We have revised the paper to include basic knowledge of the OP diffeomorphism, ODE, and flow map. This is beneficial to increase the readability of this paper for people who are not in the field of mathematics. The citations of references have also been revised accordingly.
>
> 2. We have carefully read the CMOW paper and related papers provided by the reviewer. We find the Compositional Matrix Space Model (CMSM) is very relevant to our results, so we have added a related work item, word embedding, to discuss it in the revised paper.
>
>    * CMSM is an interesting model to embed words in a way that captures order information. This advantage is also applicable to the mapping composition considered in our paper.
>    * CMSM is significantly different from our setup. CMSM considers linear mappings, which come from a finite-dimensional space. In contrast, the space of continuous mappings that we're considering is infinite-dimensional. Embedding words as nonlinear mappings would be expected to have better expressive power. However, the corresponding algorithms will be more complex.
>    * As an application of our results, we can show that CMSM also has universal approximation properties (see the added Appendix D). That is, there exists a finite set V of matrics such that any matrix can be approximated by compositing a sequence of matrics in V. We believe this result provides theoretical support for CMSM from the approximation perspective, which is absent in the literature.

---

> > ### Comment · Reviewer_gE5q · 2023-12-04
> > **Response to rebuttal**
> >
> > Thank you, these additions are helpful.

---

### Official Review · Reviewer_zLrM · 2023-11-07

**Soundness:** 3 good
**Presentation:** 3 good
**Contribution:** 2 fair
**Rating:** 5
**Confidence:** 3

**Summary:**

*I was asked to provide an emergency review for this paper after having reviewed a very similar version of this paper submitted to NeurIPS. Overall, I find that there have been minimal changes in this version and that the concerns I previously raised about the claimed link between these results and linguistic compositionality have not been addressed.*

This paper provides universal approximation results with a compositional structure. That is, they give a universal approximation result for natural classes of continuous functions where the approximator is a finite composition of functions in a finite set of atomic continuous functions (flow maps). Most of the paper is devoted to giving at the proof, which is repeatedly building up to the greatest level of detail. Specifically, the two key aspects of the proof are:
1. Decompose target function \Psi into alternating elements of two sets H1 and H2. This maps roughly onto 3.2 and directly onto 4.1/4.2 2. 2. Approximate any element of H1 or H2 with composition of flow maps in the finite vocabulary V. This maps onto 3.3

**Strengths:**

1. I am not an expert in approximation theory, but the results seem correct to the best of my knowledge. I have noted below some sources of confusion.
2. I appreciate the creative motivation for the paper and interdisciplinary aspiration: bringing together dynamical systems, approximation theory, and linguistics to better understand if and how neural networks achieve compositionality seems like a grand challenge worth pursuing.

**Weaknesses:**

The authors claim several times that their work is interesting because of a connection to the idea of compositionality in linguistics:

> Our results provide a linguistic perspective of composite mappings and suggest a cross-disciplinary study between linguistics and approximation theory

> We built an analogy between composite flow maps and words/phrases/sentences in natural languages (Table 1), which could inspire cross-disciplinary studies between approximation theory, dynamical systems, sequence modeling, and linguistics.

> Our result was inspired by the fact of finite vocabulary in natural languages, where V can be mimicked to a “vocabulary”, H1 and H2 to “phrases”, and HV to “sentences”. Our results provide a linguistic aspect for composite mappings, and we hope our findings will in turn inspire related research in linguistics.

Simply put, the results in this paper do not provide a "linguistic perspective" and are not "cross-disciplinary", and I am skeptical that they engage with linguistic ideas deeply enough to inspire research in linguistics, as suggested. Since the previous version, the authors have briefly elaborated on how their results could connect to linguistics:

>  For example, the analogies can offer novel ideas for understanding or constructing NLP models. One can embed words as functions instead of vectors in traditional models, and then the text generation problems can be converted to function approximation problems. However, constructing such models involves a number of techniques that are beyond the scope of this paper

However, I don't find this to be very concrete or convincing. I will repeat some of the points mentioned in my last review about how the ideas in this paper about how a compositional approximation result could be brought to have more linguistic significance. I believe the authors either need to address these points (especially the limitation of compactness for thinking about linguistic inputs) or they should remove any claim of linguistic applicability of their results.

## Compositionality and Relevance to Linguistics

If the authors want to claim a connection to linguistics, they should concretely discuss relevant ideas about compositionality from linguistics and formal language theory in order to ground their results. Compositionality in linguistics is the idea that a finite vocabulary of basic elements can be combined via a grammar to express an infinite range of meanings. There is a rich literature on compositionality in humans and neural networks, both from theoretical and empirical perspectives. One foundational viewpoint comes out of formal language theory, where it is studied how finite grammars can generate infinite formal languages.

Another more semantic view on compositionality is the [work of Montague]([http://wwwhomes.uni-bielefeld.de/mkracht/html/montague.pdf](http://wwwhomes.uni-bielefeld.de/mkracht/html/montague.pdf)), who argues that compositionality can be understood as an algebraic relation between the input space of strings and the output space of meanings. Meanings can be viewed as operators X → X on the world state X (i.e., discrete-time dynamical system), and we could identify each vi in the vocabulary with some meaning fi. Then one Montagovian notion of compositionality would be that concatenating vi’s is isomorphic to compositition in the meaning space. In other words, the meaning of v1 v2 v3 would be the composition of the functions f1, f2, f3. See [this paper]([https://aclanthology.org/2022.coling-1.525.pdf](https://aclanthology.org/2022.coling-1.525.pdf)) for a recent invocation of these ideas in the study of neural networks.

It seems like your results could be better interpreted if tied to this Montagovian framing. You could assume a bijection between your set of functions F and a finite vocabulary V. Then all possible strings over V defines an input space of strings, and each string v = v1 v2 v3 maps to a function f that is the composition of f1, f2, f3. From this point of view, the interpretation of your result would be that any function over a compact domain can be expressed compositionally in terms of V. This is a bit different from the standard notion of compositionality in linguistics: that the meaning of the whole input is a composition of the meanings of different parts of the input. For you, the primitive “parts” are just some finite set independent from the input. Another way of saying this is that your universal approximation construction is compositional in V, not in the input, which makes it unclear what exactly is relevant for linguistic compositionality.

However, this perspective does make me see a connection between your work and logic. One way to understand a logic is simply as a finite compositional system for expressing functions (predicates over models). From this point of view, one way to interpret your result is that general functions can be expressed in a certain logic that captures the operations in V as well as composition. Perhaps this framing might be a better description of the takeaways from your paper.

## Discrete vs. Continuous and Compactness

Additionally, my main issue was that there is a disconnect between compositional approximation of functions over compact continuous domains and functions over discrete sequences (which is what people care about in linguistics). The authors have not addressed these concerns or even mentioned this limitation. I have summarized my comments on this from the last review below.

An issue for drawing connection between these results and linguistics is is the disconnect between continuous and discrete domains. Your main theorems apply for functions over continuous domains and rely on compactness. In contrast, the type of functions most relevant in linguistics are over discrete domains: either string recognition problems (V* → {0, 1}) or string transductions (V* → V*), and is not a natural notion here.

A natural idea is to embed these discrete domains into continuous domains and apply continuous universal approximation, but this does not quite work. For an unbounded input sequence in V*, either the continuous representation will be non-compact or the precision will have to grow in the sequence length. We conclude that compactness is not achievable with bounded precision, meaning that universal approximation constructions relying on compactness will fall apart for long sequences (similar points have been made in the formal languages literature on neural networks, e.g.: [https://arxiv.org/abs/2106.16213](https://arxiv.org/abs/2106.16213)). I still find your results interesting and relevant, but given that your explicit goal is to make a connection between approximation theory and linguistics (where discrete sequences are relevant), I believe it is necessary to mention this caveat. Otherwise, some readers will not recognize that the “compact domain” condition really means your results apply with bounded length.

## Comments on Presentation

I also had several comments and suggestions on clarity from the last version that have not been clarified in this version of the paper.

You should say something briefly about what flow maps are in either 2.2 or 3 besides simply citing Arrowsmith & Place. For example, a flow map is the mapping from the initial value x0 to the state of the system after time tau, where x0 is allowed to vary. In the previous round of review, you noted you had added some clarification of this, but I couldn't find it in this version and would suggest changing 2.2 or 3 specifically.

Additionally, in some parts the notation is verbose to the point where I either recognize it could be simplified for readability or do not understand it:
- In line 88, V, V_F are the same, and F and V are distinguished only so that you can introduce a running index i over the elements of F? Just define the set one and say you will use an enumeration of the elements in the set. Also, would be more conventional to use \Sigma instead of V for the finite vocabulary from which words will be formed.
- Equation 10: Notation and indexing is unclear at first glance. Why not something like h_1 * g_1 * h_2 * g_2 … ?
- Equation 13: what is the upside down plus/minus? I like that the authors have run through the structure of the proof several times at different levels of abstraction (Section 2, 3, and 4). However, I think this could be made more explicit for the reader, with back references to the previous presentation. There could also be more through text so that the flow between sections and paragraphs is easier to understand. Additionally, it is a bit confusing that the Part 1/Part 2 structure is reflected in Section 3, but not in Part 4.

**Questions:**

In the previous round of review and paper discussion, you mentioned:

> To be honest, our manuscript is also motivated by another question: are there any other word embedding methods other than word vectors? Our theorem gives an answer and suggests embedding words as functions or dynamical systems.

Could you clarify what you meant by this?

---

> ### Author Response · Authors · 2023-11-18
> **Response to Reviewer zLrM**
>
> We thank the reviewer again for the wonderful comments and suggestions. Your knowledgeable and insightful replies give us lots of encouragement and inspiration and deserve more time to understand and think further.
>
> When submitting ICLR, we considered your previous suggestions. Either we enhance our arguments through experiments, or we delete the content related to linguistics. Since we cannot add convincing experiments in a short period, and deleting the content related to linguistics will make our theorems seem boring and unintuitive, we chose to make minor revisions.
>
> Based on further comments and suggestions from reviewers, we have revised the paper as follows.
>
> - **Linguistic perspectives and cross-disciplines**. To be precise, what this article aims to emphasize is that our results are similar to some (but not all) features of natural languages. It is natural and interesting to think about the reverse "analogy," but since we have no evidence to support such an analogy, we have revised the paper to remove the "analogy" claim. Instead, we add a paragraph to discuss the difference between our results and the compositionality concept in linguistics.
> - **Discrete vs. Continuous and Compactness**. We revised the paper to address the compactness in the conclusion section. There are differences between discrete and continuous systems, such as the difference between ResNet and neural ODE. We thank the reviewer for providing the case in linguistics. For the approximation, it is possible to extend the compact domain to unbounded domains (see [this paper](https://doi.org/10.1016/j.neunet.2021.10.001) for example). However, functions on unbounded domains should satisfy certain conditions, such as having finite integrals. We are not sure if this development is related to problems in linguistics.
> - **Notations and structures**. We revised the notations to reduce misunderstanding, and we added a subsection to introduce basic knowledge for involved concepts. In addition, we exchanged the order of Part 1/2, and the title of Sec 3/4, which make the correspondence more clear.
> - **Word embedding motivation**. The question comes to our mind when we notice the word2vec model. From a mathematical point of view, vector spaces are simple, and we were curious whether they could capture the complexity of languages. In the literature provided by the reviewer gE5q, there have been studies on embedding words as mappings, so we added a paragraph in the revised paper to discuss the word embeddings.

---

> > ### Comment · Reviewer_zLrM · 2023-11-22
> >
> > Thank you for your revisions and response.
> >
> > The reorganization of the preliminaries and context section definitely improves the readability of the paper. I also appreciate that you have made an effort to more concretely describe the claimed connection to linguistic compositionality under the revision time constraints. The connection is still somewhat tenuous ("a little similar to the compositionality in linguistics") though no longer overclaimed, which is good. I do believe that spending some more time to make the connection more precise could substantially improve the value of the paper, and maintain my score for that reason.

---

### Official Review · Reviewer_jZQe · 2023-11-16

**Soundness:** 2 fair
**Presentation:** 2 fair
**Contribution:** 2 fair
**Rating:** 3
**Confidence:** 3

**Summary:**

The paper uses an analogy between word sequences and compositions of continuous maps from analysis to obtain a compositional way of learning word embeddings in NN's.

**Strengths:**

It is nice that the mathematical theory underlying neural networks are used to instantiate a result on the application side.

**Weaknesses:**

The main problem is that the analogy upon which the paper is based fails. In linguistics, composition works along side syntactic structure. This means that each sequence of words that constitutes a sentence has a syntactic structure and that this structure defines what are the phrases within the sentence and how they are composed with each other. The paper assumes that this composition works along side words and is word by word. The phrase and sentence syntactic structures are completely ignored.  In other words, simple sequential compositions of maps does not correspond to linguistic composition.

I think the authors can remedy this by looking at the elementary fragment of a language of their choice, often taken to be English, and provide compositions alongside the phrase structures there. This latter is easily represented in a generative grammer and via Chomsky's original definitions. For instance a sentence S is generated by an NP followed by a VP: S-> NP VP.  An NP is generated by a determiner followed by a singular noun, or a plural noun on its own, NP -> Det Noun | PlNoun| ... etc etc.

I would like to strongly encourage the authors to do this.

**Questions:**

None

---

> ### Author Response · Authors · 2023-11-18
> **Response to Reviewer jZQe**
>
> We thank the reviewer for the comments and suggestions from the field of linguistics.
>
> It should be noted that the authors of this paper are not experts in the field of linguistics. The main contribution of this paper is to prove the existence of finite mapping vocabularies in the field of universal approximation. This is not an obvious result and its proof is nontrivial. We call it a linguistic perspective of mapping compositions because the vocabulary used in natural languages is also finite, which is similar to our result.
>
> As the reviewer zLrM is also concerned, the term "analogy" in our paper is misleading because "analogy" implies that the approximation properties can be brought to have more linguistic significance. This is a strong claim. We do not think it is time to answer the possibility of such an "analogy" and we keep open for any answers. As suggested by the reviewer, there is still a lot of further research needed to explore this issue, which we list as future work.
>
> Since we only aim to use some concepts in linguistics to make our results more intuitive, we have deleted the "analogy" claim in the revised paper. This avoids overclaiming our results.
>
> We would appreciate it if the reviewer could reevaluate our revised paper from the perspective of approximation. Sharing more insights and thoughts are also welcome.

---

### Meta-Review · Area_Chair_ZHEo · 2023-12-06

**Metareview:**

The paper presents a proof of a universal approximation result stating that any mapping can be represented as the composition of functions selected from a finite set of functions. The main idea of the constructive proof is to demonstrate that a sequence of flow maps derived from ODEs can be combined to approximate any given continuous mapping. The paper draws an analogy between this result and compositionality in linguistics, where sentences are build from a finite vocabulary. However, the paper does not make this connection clear enough, and do not engage sufficiently with previous work on compositionality in language. Without this, the paper has a purely theoretical results that does not have a clear enough relevance to ICLR, and therefore the paper cannot be accepted in its current form.

**Justification For Why Not Higher Score:**

The theoretical results in the paper are not relevant enough to ICLR without more clearly framing their relation to compositionality in language, or to sequence modelling more generally.

**Justification For Why Not Lower Score:**

N/A

---

### Decision · Program_Chairs · 2024-01-16

Reject